# Antiferromagnetic domain wall as spin wave polarizer and retarder

Jin Lan [1], Weichao Yu[1] & Jiang Xiao [1,2,3]

As a collective quasiparticle excitation of the magnetic order in magnetic materials, spin wave, or magnon when quantized, can propagate in both conducting and insulating materials. Like the manipulation of its optical counterpart, the ability to manipulate spin wave polarization is not only important but also fundamental for magnonics. With only one type of magnetic lattice, ferromagnets can only accommodate the right-handed circularly polarized spin wave modes, which leaves no freedom for polarization manipulation. In contrast, antiferromagnets, with two opposite magnetic sublattices, have both left and right-circular polarizations, and all linear and elliptical polarizations. Here we demonstrate theoretically and confirm by micromagnetic simulations that, in the presence of Dzyaloshinskii-Moriya interaction, an antiferromagnetic domain wall acts naturally as a spin wave polarizer or a spin wave retarder (waveplate). Our findings provide extremely simple yet flexible routes toward magnonic information processing by harnessing the polarization degree of freedom of spin wave.

[1] Department of Physics and State Key Laboratory of Surface Physics, Fudan University, Shanghai 200433, China. [2] Collaborative Innovation Center of Advanced Microstructures, Nanjing 210093, China. [3] Institute for Nanoelectronics Devices and Quantum Computing, Fudan University, Shanghai 200433, China. Jin Lan and Weichao Yu contributed equally to this work  Correspondence and requests for materials should be addressed to J.X. (email: xiaojiang@fudan.edu.cn)

Spintronics extends the physical limit of conventional electronics by harnessing the electronic spin, another intrinsic degree of freedom of an electron besides its charge[1, 2]. As a collective excitation of ordered magnetization in magnetic systems, spin wave (or magnon when quantized) carries spin angular momentum like the spin-polarized conduction electron. Different from the spin transport by conduction electrons, the propagation of spin wave does not involve physical motion of electrons. Therefore, spin wave can propagate in conducting, semiconducting, and even insulating magnetic materials without Joule heating, a troubling issue faced by the present day silicon-based information technologies[3]. The dissipation in the spin wave system is mainly due to the magnetic damping, which is much weaker than the electronic Joule heating, especially in magnetic insulators[4]. Magnonics is a discipline of realizing energy-efficient information processing that uses spin wave as its information carrier[5–7]. Besides its low-dissipation feature, the wave nature and the wide working frequency range of spin wave make magnonics a promising candidate of upcoming beyond-CMOS (complementary metal oxide semiconductor) computing technology.

Similar to the intrinsic spin property of elementary particles, polarization is an intrinsic property of wave-like (quasi-)particles such as photons, phonons, and magnons. It is more natural to encode information in the polarization degree of freedom than other degrees of freedom such as amplitude or phase. For instance, photon polarization has been widely used in encoding both classical and quantum information[8], and manipulation of photon polarization is essential for applications in photonics[9]. The phonon polarization has also been used in encoding acoustic information for realizing phononic logic gates[10, 11], and for controlling magnetic domain walls[12]. In contrast, most magnonic devices proposed or realized so far mainly use the spin wave amplitude[13–18] or phase[19–23] to encode information, and spin wave polarization is rarely used except in very few cases[24]. The lack of usage of polarization in magnonics has reasons. With only one type of magnetic lattice, ferromagnets can only accommodate the right-circular polarization, hence there is simply no freedom in polarization manipulation. This situation is similar to the case of half-metal, which has only one spin spieces[25]. In other words, ferromagnet is a half-metal for spin wave. To overcome this disadvantage, a straightforward approach is to use antiferromagnets in manipulating spin wave polarization. In antiferromagnets, due to the two opposite magnetic sublattices, spin wave polarization has complete freedom as the photon polarization. Thus, antiferromagnet is a normal metal for spin wave with all possible polarizations. In comparison with the ferromagnets, besides gaining the full freedom in polarization, antiferromagnets also have numerous other advantages, such as the much higher operating frequency (up to THz) and having no stray field, etc[26, 27].

Due to these merits, antiferromagnet is regarded as a much better platform for magnonics than ferromagnet[24, 28–30]. In order to unleash the full power of antiferromagnets in magnonics, however, it is highly desirable to have a simple yet efficient way in manipulating spin wave polarization, ideally in a similar fashion as in its optical counterpart.

In general, to be able to manipulate polarization with full flexibility, two basic devices are indispensable, i.e., the polarizer and retarder (waveplate)[9, 31]. The former is to create a particular linear polarization, and the latter is to realize the conversion between circular and linear polarizations. The actual realization of these two building blocks relies on the types of the (quasi-)particle in question. For instance, for photon, an array of parallel metallic wires functions as an optical polarizer[9], and a block of birefringent material with polarization-dependent refraction indices acts as an optical waveplate[31].

Here, we show theoretically and confirm by micromagnetic simulations that, utilizing the Dzyaloshinskii-Moriya interaction (DMI) existing in the symmetry-broken systems[32, 33] an antiferromagnetic domain wall serves as a spin wave polarizer at low frequencies and a retarder at high frequencies. Due to the extreme simplicity and tunability of a magnetic domain wall structure, the manipulation of spin wave polarization in magnonics becomes not only possible but also simpler and more flexible than its optical counterpart.

## Results

**Model**. We consider a domain wall structure in an one-dimensional antiferromagnetic wire along $\hat{\mathbf{x}}$ direction as shown in Fig. 1, where the *red/blue* arrows denote the magnetization direction $\mathbf{m}_{1,2}$ for the two magnetic sublattices of an antiferromagnetic domain wall. The domain wall profile is taken as Bloch type, where the magnetization rotation plane ($y$–$z$ plane) is perpendicular to the domain wall direction ($\hat{\mathbf{x}}$). The magnetization dynamics in the antiferromagnetic wire can be described by two coupled Landau–Lifshitz–Gilbert (LLG) equations for each sublattice[34, 35],

$$\dot{\mathbf{m}}_i(\mathbf{r}, t) = -\gamma \mathbf{m}_i(\mathbf{r}, t) \times \mathbf{H}_i^{\text{eff}} + \alpha \mathbf{m}_i(\mathbf{r}, t) \times \dot{\mathbf{m}}_i(\mathbf{r}, t), \quad (1)$$

where $i = 1, 2$ denote the two sublattices, $\gamma$ is the gyromagnetic ratio, $\alpha$ is the Gilbert damping constant. Here $\gamma \mathbf{H}_i^{\text{eff}}(\mathbf{r}, t) = K m_i^z \hat{\mathbf{z}} + A \nabla^2 \mathbf{m}_i + D \nabla \times \mathbf{m}_i - J \mathbf{m}_{\bar{i}}$ (with $\bar{1} = 2$ and $\bar{2} = 1$) is the effective magnetic field acting locally on sublattice $\mathbf{m}_i$, where $K$ is the easy-axis anisotropy along $\hat{\mathbf{z}}$, $A$ and $D$ are the Heisenberg and Dzyaloshinskii-Moriya (DM) exchange coupling constant within each sublattice, and $J$ is the exchange coupling constant between two sublattices. The Heisenberg exchange coupling tends to align neighboring magnetization in parallel. While the DM exchange coupling, existing in magnetic materials or structures lack of

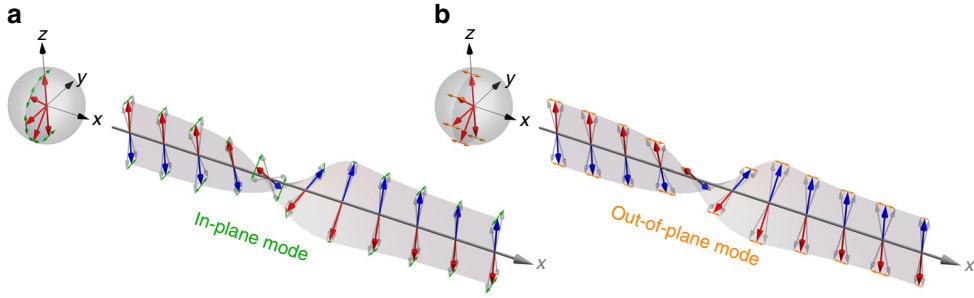

**Fig. 1** Schematics of an antiferromagnetic domain wall with its in-plane and out-of-plane modes. The domain wall profile is indicated by the *thick red/blue arrows* for the two sublattices. The Bloch sphere on the *left* shows that the magnetization $\mathbf{m}_1$ (*red arrow*) in a domain wall traces a longitude on a Bloch sphere in the $y$–$z$ plane from *north pole* to *south pole*. The *green* (at **a**) and *orange* (at **b**) *arrows* indicate the the in-plane (at **a**) and out-of-plane (at **b**) spin wave excitations upon a Bloch type antiferromagnetic domain wall. The in-plane and out-of-plane modes oscillate along longitude and latitude on the Bloch sphere, and they are connected to linear $y$-polarization and $x$-polarization inside domains (where $\mathbf{m}_i \| \hat{\mathbf{z}}$), respectively

spatial/inversion symmetry[32, 33], prefers magnetization to rotate counter-clockwise about an axis (x-axis in our case) determined by the symmetry-broken direction.

We denote the static profile of the antiferromagnetic domain wall along x-axis as $\mathbf{m}_1^0(x) = -\mathbf{m}_2^0(x)$, and the stagger order $\mathbf{n}_0(x) \equiv (\mathbf{m}_1^0 - \mathbf{m}_2^0)/2 = (\sin\theta_0\cos\phi_0, \sin\theta_0\sin\phi_0, \cos\theta_0)$, where $\theta_0(x)$ and $\phi_0(x)$ are the polar and azimuthal angle of $\mathbf{n}_0$ with respect to $\hat{\mathbf{z}}$ (see Fig. 1). No matter DMI is present or not, an antiferromagnetic domain wall in Fig. 1 always takes the Walker type profile like its ferromagnetic counterpart with $\theta_0(x) = -2\arctan[\exp(x/\Delta)]$ and $\phi_0 = \text{const}$.[16, 30, 36], where $\Delta = \sqrt{A/K}$ is the characteristic domain wall width. (See Supplementary Fig. 1, Supplementary Note 1 and Supplementary Movie 1). The effect of DMI is to determine the domain wall type and chirality: when DMI is absent, the domain wall is free to take either the Bloch type ($\phi_0 = \pi/2$) or Néel type ($\phi_0 = 0$) configuration, or any mixture of the two with $0 < \phi_0 < \pi/2$; in the presence of DMI, the domain wall becomes chiral[37, 38] and is pinned to the Bloch type as shown in Fig. 1.

With the static profile $\mathbf{m}_i^0(x)$, let $\mathbf{m}_i(x,t) = \mathbf{m}_i^0(x) + \delta\mathbf{m}_i(x,t)$ and $\delta\mathbf{m}_i(x,t) = m_i^\theta(x,t)\hat{\mathbf{e}}_\theta + m_i^\phi(x,t)\hat{\mathbf{e}}_\phi$ be the dynamical spin wave excitation upon the static $\mathbf{m}_i^0(x)$, where $\hat{\mathbf{e}}_\theta$ and $\hat{\mathbf{e}}_\phi$ are the local transverse (polar and azimuthal) directions with respect to $\mathbf{n}_0(x)$. Since $\hat{\mathbf{e}}_\theta$ lies in the magnetization rotation plane (the y–z plane), we call the excitation in $\hat{\mathbf{e}}_\theta$ the in-plane mode. Similarly, the excitation in $\hat{\mathbf{e}}_\phi$ is perpendicular to the rotation plane and is called the out-of-plane mode (see Fig. 1). By eliminating the static profile from the LLG Eq. (1), the spin wave dynamics is governed by the following linearized LLG equations:

$$\dot{m}_{\mp}^\phi = -\left[-A\frac{\partial^2}{\partial x^2} + V_K(x) + J \mp J\right]m_\pm^\theta, \quad (2)$$

$$\dot{m}_\pm^\theta = +\left[-A\frac{\partial^2}{\partial x^2} + V_K(x) + J \pm J + V_D(x)\right]m_\mp^\phi, \quad (3)$$

where $m_\pm^{\phi,\theta} = m_1^{\phi,\theta} \pm m_2^{\phi,\theta}$. The effects of the inhomogeneous domain wall texture are transformed into two effective potentials $V_K(x)$ and $V_D(x)$[16, 36]: $V_K(x) = K[1 - 2\text{sech}^2(x/\Delta)]$ arises due to the easy-axis anisotropy along $\hat{\mathbf{z}}$, and $V_D(x) = (D/\Delta)\text{sech}(x/\Delta)$ is due to the combined action of DMI and the inhomogeneous magnetic texture. Equations (2) and (3) can be regrouped into two independent sets for $(m_+^\phi, m_-^\theta)$ and $(m_-^\phi, m_+^\theta)$, whose solutions are two linearly polarized spin wave modes oscillating in the (in-plane) $\hat{\mathbf{e}}_\theta$-direction and (out-of-plane) $\hat{\mathbf{e}}_\phi$-direction, respectively, as depicted in Fig. 1. Deep inside each domain (where $\mathbf{n}_0 \| \pm\hat{\mathbf{z}}$), the two transverse directions $\hat{\mathbf{e}}_\theta$ and $\hat{\mathbf{e}}_\phi$ coincide with $\hat{\mathbf{y}}$ and $\hat{\mathbf{x}}$, therefore we also call the in-plane (out-of-plane) modes y-polarized (x-polarized) modes.

The influences of a domain wall on spin wave propagation are all captured by the effective potentials $V_K$ and $V_D$. In the short wavelength limit (WKB approximation), using Eqs. (2) and (3), we may define the local dispersion at position x for chosen polarization and wavevector k:

in-plane : $\quad \omega^{\text{IP}}(k,x) = \omega_\theta(k,x)$
$$= \sqrt{[2J + Ak^2 + V_K(x) + V_D(x)][Ak^2 + V_K(x)]}, \quad (4)$$

out-of-plane : $\quad \omega^{\text{OP}}(k,x) = \omega_\phi(k,x)$
$$= \sqrt{[2J + Ak^2 + V_K(x)][Ak^2 + V_K(x) + V_D(x)]}, \quad (5)$$

which gives the local spin wave gap as $\omega_G^{\text{IP/OP}}(x) \equiv \omega^{\text{IP/OP}}(0,x)$, below which no in-plane/out-of-plane mode is allowed to propagate. In the absence of DMI ($V_D = 0$), the in-plane and out-of-plane modes are degenerate ($\omega^{\text{IP}} = \omega^{\text{OP}}$). This degeneracy is lifted by the introduction of DMI (the $V_D$ term in Eqs. (4) and (5)), with which the local spin wave gap for the out-of-plane modes is higher than that for the in-plane modes inside the domain wall

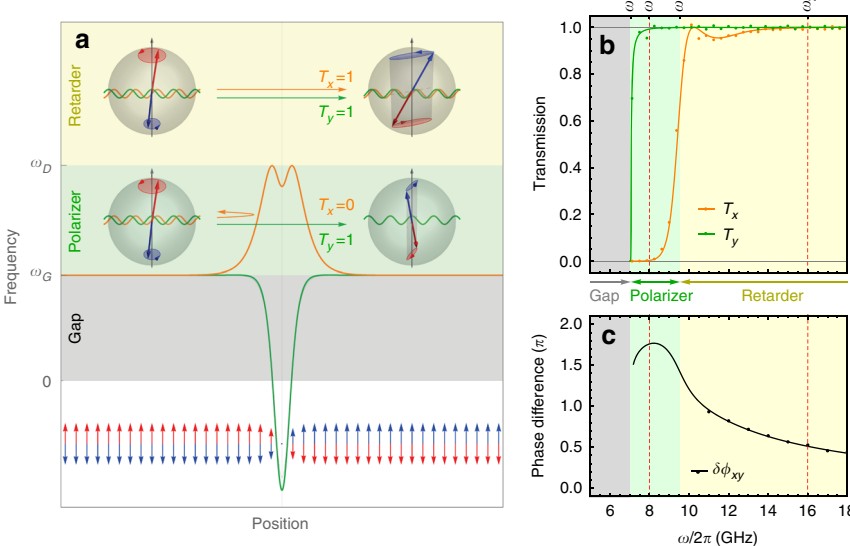

**Fig. 2** Polarization-dependent spin wave scattering by an antiferromagnetic domain wall. **a** The schematic picture of the effective potentials (the square of the local spin wave gap) in an antiferromagnetic domain wall for the x-polarized and y-polarized spin wave modes. The effective potential for y-polarization (the *green curve*) is nearly reflectionless, hence transmission probability $T_y \simeq 1$ for $\omega > \omega_G$; the effective potential for x-polarization (the *orange curve*) is a barrier at the domain wall due to $V_D$, hence transmission probability $T_x \simeq 0$ for $\omega_G < \omega < \omega_D$ and $T_x \simeq 1$ for $\omega > \omega_D$. The different effective potentials for x and y polarizations also give rise to polarization-dependent phase delays. Therefore, the antiferromagnetic domain wall is a spin wave polarizer in the frequency range between $\omega_G$ and $\omega_D$, and a spin wave retarder in the frequency range above $\omega_D$. **b** The transmission probability $T_{x,y}$ for the linear x-polarized and y-polarized spin wave modes. **c** The relative phase accumulation $\delta\phi_{xy} = \phi_y - \phi_x$ between y-polarization and x-polarization across the domain wall. In **b**, **c**, the solid curves are calculated from the Green's function method and the dots are obtained from the micromagnetic simulations. Frequencies $\omega_1/2\pi = 8$ GHz and $\omega_2/2\pi = 16.2$ GHz denote the position of the working frequencies used in the simulations for spin wave polarizers in Fig. 3 and spin wave quarter-wave plates (retarders) in Fig. 4, respectively

$(\omega_G^{OP}(x) \geq \omega_G^{IP}(x))$ (see the plots of the local spin wave gap in Fig. 2a). Deep inside each domain ($|x| \gg \Delta$, $V_K \to K$, $V_D \to 0$), no matter whether DMI is present or not, both dispersions in Eqs. (4) and (5) reduce to the standard antiferromagnetic dispersion $\omega_0(k) = \sqrt{(2J + K + Ak^2)(K + Ak^2)}$ with a spin wave gap $\omega_G = \sqrt{(2J + K)K}$[34, 39]. It is seen that the dispersions inside domains is not modified by DMI, this is because that DMI modifies the spin wave dispersion with a linear term in wavevector only when the wavevector has component (anti-)parallel to the magnetization direction[40], and in our case the spin wave wavevector ($\hat{x}$) is perpendicular to the magnetization direction ($\hat{z}$).

**Polarization-dependent scattering.** In the absence of DMI ($D = V_D = 0$), the dispersions and the local spin wave gaps for in-plane and out-of-plane polarization in Eqs. (4) and (5) are identical, thus the scattering behavior of spin waves by the domain wall is independent of polarization. Furthermore, the potential $V_K(x)$ is the well known reflectionless Pöschl–Teller type potential well[16, 18, 36, 41]. Consequently, regardless of its polarization, the incident spin wave experiences no reflection by the domain wall, and only accumulate a common phase delay. However, when DMI is present ($D$, $V_D \neq 0$), the dispersions and local spin wave gaps for the in-plane and out-of-plane modes are different. Considering that the inter-sublattice exchange coupling $J$ is the dominating energy scale in antiferromagnet, the in-plane gap $\omega_G^{IP}$ is barely

affected by $V_D$ ($\ll 2J$). On the contrary, the out-of-plane gap $\omega_G^{OP}$ is elevated by $V_D > 0$ and reaches its maximum value $\omega_D$ inside the domain wall. This effect of DMI in Eqs. (4) and (5) is equivalent to an effective hard-axis anisotropy along the out-of-plane ($\hat{x}$) direction inside the domain wall[41, 42], which suppresses the out-of-plane excitation. Consequently, the domain wall can still be regarded as reflectionless for the in-plane $y$-polarization, but becomes a potential barrier of height $\omega_D$ for the out-of-plane $x$-polarization (see Fig. 2a). Therefore, the $x$- and $y$-polarized modes are scattered differently: at low frequencies ($\omega_G < \omega < \omega_D$), the $y$-polarization experiences no (or little) reflection but the $x$-polarization hits a potential barrier and is strongly reflected; At higher frequencies ($\omega > \omega_D$), both polarization transmit almost perfectly, but the $x$- and $y$-polarization no longer share the same phase delay.

The polarization-dependence of the spin wave scattering behaviors by the domain wall are confirmed by micromagnetic simulations based on the LLG Eq. (1) and Green's function calculations based on linearized LLG Eqs. (4) and (5). (See Supplementary Note 3 for details) Using these methods, we calculate the spin wave transmission probabilities $T_{x,y}$ through the domain wall for the linear $x$- and $y$-polarization, as well as their relative phase accumulation $\delta\phi_{xy} = \phi_y - \phi_x$, as presented in Fig. 2b, c. It is seen that the $y$-polarization transmit almost perfectly ($T_y \simeq 1$) for all frequencies above the spin wave gap ($\omega > \omega_G$). In contrast, because of the effective potential barrier, the $x$-polarization experiences total reflection ($T_y \simeq 0$) at low

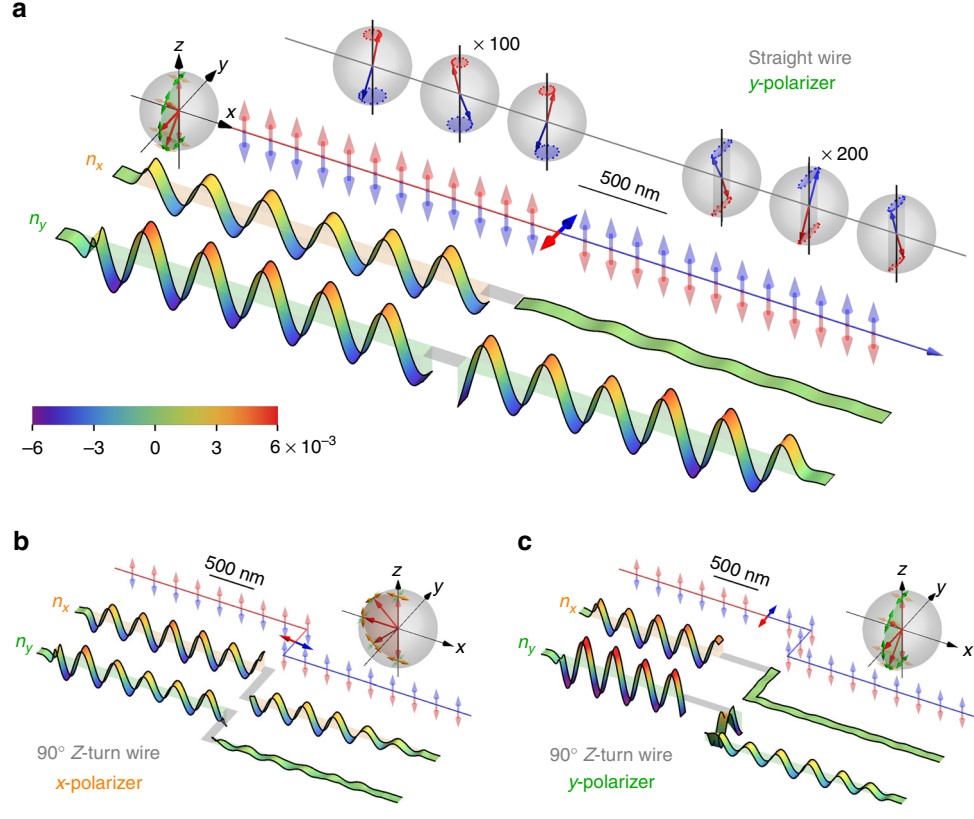

**Fig. 3** Micromagnetic simulation of spin wave polarizers. **a** An antiferromagnetic domain wall in a straight wire works as a $y$-polarizer. The four rows from top to bottom are: (i) spin wave excitations depicted on Bloch spheres, with the spin wave amplitude exaggerated by 100 (200) times at the left (right) side. (ii) the static magnetization profile $\mathbf{m}_1$ (red) and $\mathbf{m}_2$ (blue). The Bloch sphere on the left shows that the magnetization in the domain wall rotates in the $y$–$z$ plane with $x$-polarization suppressed, (iii–iv) the instantaneous wave form of the $x$ and $y$ component of the stagger order $n_{x,y}$ at a selected time. **b** An $x$-polarizer when the domain wall is at the center segment of a Z-turn wire, and the magnetization in the damain wall rotates in $x$–$z$ plane with $y$-polarization suppressed. **c** A $y$-polarizer when the domain wall is at the left segment of a Z-turn wire, and the magnetization rotates in $y$–$z$ plane with $x$-polarization suppressed. For all figures, the spin wave injected from left has circular polarization and frequency $\omega/2\pi = 8$GHz ($\omega_1$ in Fig. 2b, c). The damping coefficient in the simulation is $\alpha = 5 \times 10^{-4}$. The wave form within the domain wall region is omitted

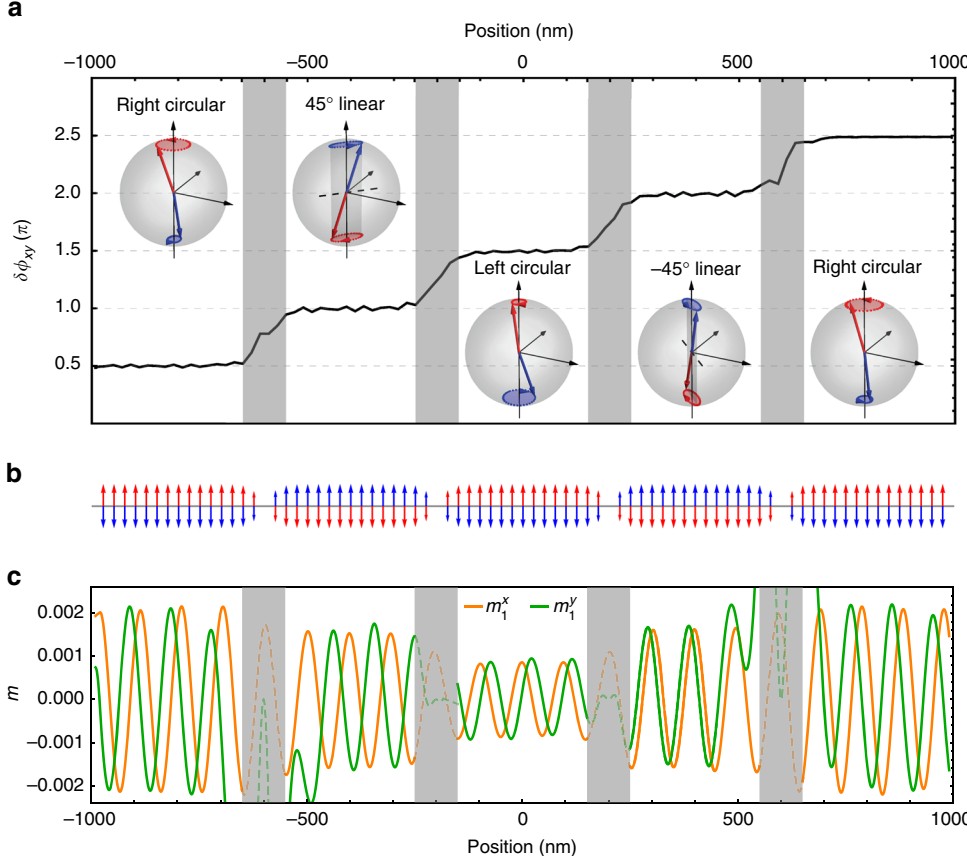

**Fig. 4** Micromagnetic simulation of a series of spin wave retarders. At working frequency $\omega/2\pi = \omega_2/2\pi = 16.2$ GHz, each domain wall is a quarter-wave plate. The right-circular spin waves are injected from the left side. **a** The relative phase accumulation $\delta\phi_{xy}$ between the $x$-polarization and $y$-polarization components as function of position when the spin wave passes through four domain walls. The time traces of magnetization on a Bloch sphere in each domain (amplitudes exaggerated by 150 times) are shown in the insets. **b** The static domain wall profile $m_1^z(x)$ (red) and $m_2^z(x)$ (blue). **c** The wave forms of $m_1^x$ and $m_1^y$ at a given time, showing that the relative phase changes from domain to domain. In this simulation, we artificially set the damping coefficient to be zero to show that the retarder (domain wall) itself is (nearly) reflectionless and does not influence the amplitude of spin waves

frequencies ($\omega < \omega_D$). Above the barrier ($\omega > \omega_D$), however, the $x$-polarization experiences little reflection ($T_y \simeq 1$). Because of the different effective potentials experienced at the domain wall, the $x$- and $y$-polarizations develop a relative phase accumulation $\delta\phi_{xy}$, which is frequency-dependent as shown in Fig. 2c. All these behaviors are in perfect agreement with the qualitative understanding shown in Fig. 2a. Utilizing these particular spin wave scattering properties, an antiferromagnetic domain wall can be used as a spin wave polarizer and retarder (or waveplate) as discussed in the following.

**Spin wave polarizer.** In the low frequency regime ($\omega_G < \omega < \omega_D$), because the $x$-polarized spin wave is totally reflected, an antiferromagnetic domain wall is naturally a $y$-polarizer for spin wave. The functionality of the spin wave polarizer is confirmed by a micromagnetic simulation on a straight antiferromagnetic wire shown in Fig. 3a: when a circularly polarized spin wave, consisting of $x$- and $y$-polarization of equal amplitude, is injected from the left side of the domain wall, only the $y$-polarization transmits, indicating that the domain wall is a $y$-polarizer for spin wave.

It appears that the polarizer demonstrated in Fig. 3a can only be a $y$-polarizer. In fact, an $x$-polarizer can be realized simply by using a 90° Z-turn wire as shown in Fig. 3b, where the domain wall is located at the central segment. Because the domain wall is now along the $y$ direction and the static magnetization rotates from $+\hat{z}$ to $-\hat{z}$ in the $x$–$z$ plane (rather than the $y$–$z$ plane in the straight wire case in Fig. 3a), roles of $x$-polarization and $y$-

polarization interchange. Therefore, a domain wall located at the central segment of a Z-turn wire only allows the $x$-polarization to pass (instead of the $y$-polarization in the straight wire case), hence it is an $x$-polarizer as demonstrated in Fig. 3b. Actually, a Z-turn wire can work as either $x$- or $y$-polarizer depending on the location of the domain wall: an $x$-polarizer when the wall is at the center segment (Fig. 3b), and a $y$-polarizer when the wall is at either the left (Fig. 3c) or right segment.

**Spin wave retarder.** In the high frequency regime ($\omega > \omega_D$), both $x$- and $y$-polarized spin wave modes transmit almost perfectly through the domain wall, but they accumulate different phases. As a result of this relative phase delay, an antiferromagnetic domain wall functions as a spin wave retarder (or a waveplate). As shown in Fig. 2c, the relative phase delay depends on the spin wave frequency. By choosing the working frequency at $\omega/2\pi = \omega_2/2\pi = 16.2$ GHz, the relative phase delay between the $x$- and $y$-polarization $\delta\phi_{xy}(\omega_2) = \pi/2$, hence the antiferromagnetic domain wall is specifically a quarter-wave plate. Figure 4 shows a micromagnetic simulation of a right-circularly polarized spin wave passing through four consecutive domain walls, each of which is a quarter-wave plate. As shown in the top panel, the relative phase delay $\delta\phi_{xy}(x)$ increases by $\pi/2$ across each domain wall. By passing through these domain walls, the right-circular polarized spin wave is converted into a 45°-linear mode, a left-circular mode, a $-45°$-linear mode, and back into a right-circular mode, successively.

## Discussion

The antiferromagnetic domain wall based spin wave polarizer/retarder mimics its optical counterpart in many aspects. For example, the spin wave polarizer follows the Malus's law[9] as the optical polarizer, i.e., the spin wave intensity is reduced by $\cos^2\theta$ with $\theta$ the angle between the injecting polarization and the polarizer direction. Owing to the relative phase delay, an antiferromagnetic domain wall with DMI can be regarded as a birefringent material for spin wave. This birefringence brought by magnetic texture here only occurs within the domain wall region, thus it is drastically different from the birefringence caused by the biaxial anisotropy with the polarization degeneracy lifted in the whole-antiferromagnetic material[43]. In addition, both polarizing and retarding effects are independent of propagation direction, or equivalently, an up-to-down domain wall and a down-to-up domain wall are the same in polarizing or retarding. All these features enable us to construct magnonic circuits in a similar fashion as the well-established optical circuits[44]. Beyond the capability of its optical counterpart, due to the extreme tunability of magnetic textures, e.g., creating/annihilating or moving a domain wall, the magnonic circuit built upon magnetic domain walls can be modulated with even greater flexibility[16].

As demonstrated above in Fig. 3, a simple geometric modification (from a straight to a Z-turn wire) can change the polarizing behavior. This is because that the spin wave polarization is characterized in the spin space, and it does not change when the propagation takes turns in real space. However, the rotation plane of the magnetization of a domain wall depends on the domain wall direction in real space. Therefore, the polarization component allowed for transmission depends on the domain wall direction as seen in Fig. 3. Based on this principle, a linear polarizer along arbitrary direction can be realized straightforwardly using a wire with a turn of appropriate angle.

The polarizing and retarding effects sustains at smaller DMI strength $D$ (See Supplementary Fig. 2 and Supplementary Note 2). For fixed material parameters ($K$, $A$, $J$, $D$ etc), the working frequency range for the spin wave polarizer and retarder are different, i.e., a polarizer for $\omega < \omega_D$ and a retarder for $\omega > \omega_D$. However, by modulating the material parameters (such as tuning the DMI strength as in refs. [45, 46]), it is possible to have the spin wave polarizer and retarder functioning in the same frequency range. Henceforth, multiple polarizers (of different polarization direction) and retarders (of different phase delay) can be assembled into one magnonic circuit to realize more complex spin wave manipulations.

The DMI has two different forms, the bulk and the interfacial form[47], where the former is caused by the bulk inversion symmetry breaking, such as in MnSi[48] and other B20 compounds[49], and the latter is due to the interfacial inversion symmetry breaking, such as in Co/Pt or Co/Ni bilayer[50, 51]. Here, we adopt the bulk form DMI to demonstrate the working principle of the spin wave polarizer and retarder. However, the materials or structures with interfacial form DMI would work in almost the same manner. The only difference is that interfacial DMI prefers a Néel type domain wall rather than a Bloch type for the bulk DMI[37, 38] and the roles of in-plane and out-of-plane modes interchange. Besides the inhomogeneous DMI discussed above, there is also homogenous DMI that makes the antiferromagnet a weak ferromagnet by slightly misaligning the magnetizations in two sublattices[32, 33]. However, such a misalignment is extremely small because of the dominating antiferromagnetic coupling, thus is ignored in our discussions.

The working principles for the spin wave polarizer and retarder work for both real antiferromagnet and artificial synthetic antiferromagnet, where the latter consists of two antiferromagnetically coupled ferromagnetic wires[52, 53]. The domain wall structures have been observed in both types of antiferromagnetic materials[53–55], and can be effective controlled by using the exchange bias effects[30, 56–60]. Experimentally, it should be more straightforward to realize the spin wave polarizer and retarder proposed here using the synthetic antiferromagnetic wires used for the racetrack memory[53].

In conclusion, we demonstrated that the antiferromagnetic domain wall with DMI naturally functions as a spin wave polarizer at low frequency, and a spin wave retarder (waveplate) at high frequency. This extremely simple design enables all possible polarization manipulations for antiferromagnetic spin waves. Our findings greatly enrich the possibility of magnonic information processing by harnessing the polarization degree of freedom of spin wave.

## Methods

**Micromagnetic simulations**. The micromagnetic simulations are performed in COMSOL Multiphysics, where the LLG equation is transformed into weak form by using the mathematical module and solved by the generalized-alpha method. The synthetic antiferromagnet wire is composed of two ferromagnetic YIG wires with the following parameters:[16, 18, 36] the easy-axis anisotropy $K/\gamma = 3.88 \times 10^4$ A m$^{-1}$, the intra-sublattice Heisenberg exchange constant $A/\gamma = 3.28 \times 10^{-11}$ A m, the saturation magnetization $M_s = 1.94 \times 10^5$ A m$^{-1}$, the gyromagnetic ratio $\gamma = 2.21 \times 10^5$ rad m A$^{-1}$ s$^{-1}$, and the DMI constant $D/\gamma = 3 \times 10^{-3}$ A. The inter-sublattice antiferromagnetic exchange coupling between two ferromagnetic wires is $J/\gamma = 5 \times 10^5$ A m$^{-1}$. The dipolar interaction is neglected for this antiferromagnetic environment. To obtain the transmission probability, we first verify that the spin wave perfectly transmits for $D = 0$, then the transmission probability for a chosen polarization is extracted by taking the ratio between the spin wave amplitude for (i) $D = 0$ and (ii) $D \neq 0$ at the location 500 nm away from the domain wall.

**Data availability**. The data that support the findings of this study are available from the corresponding authors on request.

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

## Acknowledgements

This work was supported by the National Natural Science Foundation of China under Grant No. 11474065, National Key Research Program of China under Grant No. 2016YFA0300702, and National Basic Research Program of China under Grant No. 2014CB921600. J.L. is also supported by the China Postdoctoral Science Foundation under Grant No. KLH1512074 and No. KLH1512087. J.L. is grateful to Ran Cheng at Carnegie Mellon University for help with preparing figures.

## Author contributions

J.L. did the analytical calculations. W.Y. did the micromagnetic simulations. J.L. and W.Y. contributed equally to the work. J.X. planned and supervised the study. All authors discussed the results and worked on the manuscript.

## Additional information

**Competing interests:** Fudan University filed two Chinese Patent Applications No. 2016112619740, No. 201611261990X based on the described technologies.

