## [Peer Review File · Nature Communications]

Reviewers' Comments:

Reviewer #1 (Remarks to the Author):

Lan et al. reports a theoretical study on the spin-wave (SW) propagation through an antiferromagnetic domain wall (DW), which is subject to the Dzyaloshinskii-Moriya interaction (DMI). The authors find that the in-plane and out-of-plane SW modes experience different SW gap and thus different SW scattering, which is induced by a combined effect of DMI and DW texture. As a result, when SW passes through DMI-DW, the SW polarization changes for an intermediate SW frequency and the SW phase changes for a higher frequency. These findings lead to a conclusion that an antiferromagnetic DW subject to DMI acts as a SW polarizer or a SW retarder, depending on the SW frequency. This conclusion is supported by SW theory and micromagnetic simulation, which have been well established in the SW community.

Magnonics, which aims to utilize SWs as information carriers, is currently one of extensively studied subjects in spintronics/magnetism communities as it offers low-power computing due to negligible Joule heating and nonlinear computing due to its wave characteristics. As described in the introduction, most magnonic devices up to now have focused on the control of SW amplitude or phase, but largely ignored the possibility to use the SW polarization. As proven in spintronics and photonics, the control of relevant polarization is essential for various applications. In this respect, the idea proposed in this work is attractive and very stimulating. The manuscript is well written so that even nonspecialists may get the point of how the proposed idea works. Also the proposed idea seems to work for synthetic antiferromagnets which are easy to fabricate in lab so that I expect this work will immediately motivate experimental groups.

On the overall, I find this work is very interesting and will attract considerable attention from magnonics/spintronics/magnetism communities. Thus I support the publication of this work in Nature Communications once the authors have a chance to consider the following minor comments.

1. At the end of the 1st paragraph in page 4, it is written that "... in-plane (out-of-plane) modes x-polarized (y-polarized) modes". It seems that x and y may need to be interchanged.
2. The authors mainly discuss the action of DW on SWs. It would be nice to discuss about the reaction: how the change in the SW polarization affects the DW? Is there any relevant magnonic torque enabling DW motion?

Reviewer #2 (Remarks to the Author):

The authors claim to use a domain wall as an "optical" element, a polarizer or a phase plate for magnons. This goal is achieved by introducing inhomogeneous DMI (Lifshitz invariant) into the system Hamiltonian and analysing magnon spectra and dispersion.

Within the sine-Gordon model, which is usually used for FM and AF with uniaxial anisotropy, the domain wall does not scatter magnons (produces non-scattering potential for magnons). Any additional upgrading of the model (e.g., potential corresponding to tetragonal anisotropy) automatically gives rise to the scattering domain wall. If, in addition, this scattering is spin-dependent, then, the DW can have the properties of spin polarizer. Hence the idea is a clear one in principle, albeit not very innovative.

They start from LLG equations for each of magnetic sublattice and analyse excitations over the given equilibrium profile of the DW, analytically, by micromagnetic calc., and by what they claim to be a Green function technique.

There is however inconsistencies within their treatment. The authors consider the domain wall between two (equivalent? they do not explain) 180 domains. So, they implicitly assume that the ground state is homogeneous up/down long z axis. Moreover, as an equilibrium profile of the DW they use ansatz for AF without DMI.

The system with such a DMI is apt to formation of spiral and skyrmions. Homogeneous ground state (if any) can exist in a certain range of parameters which can be roughly estimated as follows. The system has two important lengthscales: the domain wall width $\sqrt{A/K}$ and a period of spiral A/D . Dimensionless parameter AK/D^2 separates two regimes. At $AK/D^2 \gg 1$ the ground state is almost homogeneous, and DMI modifies the DW and can (I am not sure) modify spectra. However, two homogeneous (up/down) states are equivalent in the limit of vanishingly small DMI.

At $AK/D^2 \ll 1$ the ground state is a spiral (or skyrmion). If there are two equivalent spiral states (I did not checked), then, the DW separates these states and its structure is different from the DW for 180 AF. In this case the author's approach is not valid at all.

The authors do not give any analysis of the parameters or validity of their model. However, at the end of the paper (Methods) give some numbers from which I deduce that they consider the

most complicated case $AK/D^2=1$ in which the ground state is a spiral and spiral period is equal to the domain wall width. Analytical ansatz used for the DW profile is not valid.

The authors also consider DMI as correction (which is not justified in the text) and assume that DMI "acts" only within the DW (because DMI is related with space derivative).

In a homogeneous phase DMI modifies magnons spectrum, which should include linear in k term. So, I cannot agree with the last statement in the paragraph below Eq.(3). Moreover, in a homogeneous state DMI modifies the profile of the DW, so, again, analytical ansatz used for the DW profile is approximate or not valid in the presence of DMI.

As a third comment, the authors consider DMI of the special type (inhomogeneous, Lifshits-like). The broken space-reversal symmetry allows not only inhomogeneous DMI, it can also allow homogeneous DMI. Homogeneous DMI means that the system is a weak AF, i.e., has nonzero magnetization in the initial state. This also means that magnon modes with different spins are nondegenerate from the beginning, DW also has nonzero magnetization and this can influence substantially on the polarising properties of the DW. I understand, that homogeneous DMI makes the model more complicated, but authors should explain explicitly why they disregard homogeneous DMI.

As a whole the idea is not, in my opinion, innovative enough (it has been around) to be considered for publication in Nature Communications. The many inconsistencies that are present in their analysis also brings the work to a quality that should be remedied. However, I would emphasize that even when these issues are addressed properly, I do not find the general claims (which some I question) of high enough interest for publication in Nature Communications.

Reviewer #3 (Remarks to the Author):

This is an interesting, novel and generally correct manuscript coming from one of the theoretical leaders in the field.

The authors propose to use polarization of spin waves in a dielectric antiferromagnetic (AFM) waveguide as a new way for information coding in beyond-CMOS AFM-based information processing systems, and demonstrate, that in the presence of DMI, an AFM domain wall can work as a spin wave polarizer or as a waveplate.

The main text of the manuscript is clearly written and will be accessible to a wide audience of the Nature Communications.

However, in the opinion of this Referee, the abstract and the introduction to the manuscript contain several incorrect or/and ambiguous statements, and need to be rewritten.

1. In the first sentence of the abstract the authors write :

"...spin wave, or magnon when quantized, can propagate in both conducting and insulating materials without Joule heating.

The text would become much clearer if the authors immediately mention that for spin wave Joule losses are simply replaced by losses associated with the Gilbert damping of propagating spin waves.

2. The authors, first, state that they like AFM materials for polarization manipulation, because, in contrast with ferromagnets (FM), the time-reversal symmetry is restored in AFM.

Then, they use AFM with DMI, where time-reversal symmetry is broken again by DMI.

The general relation between the polarization manipulation for signal processing and the time-reversal symmetry should be explained in a correct and consistent way.

3. The authors write in the introduction: "...polarization is an intrinsic property of wave-like (quasi-)particles such as light, acoustic wave, and spin wave."

It seems to me, that "wave-like (quasi-)particles" are not "light, acoustic wave, and spin wave", but photons, phonons and magnons.

4. The referencing of the previous work in the field is inadequate.

The authors should mention at least some of the work of their competitors in the same field:

1. Gomonay, H. V. & Loktev, V. M. Spin transfer and current-induced switching in antiferromagnets. *Physical Review B* 81, 144427 (2010).

2. Gomonay, E. V. & Loktev, V. M. Spintronics of antiferromagnetic systems (review article).

Low Temp. Phys. 40, 17–35 (2014).

3. Khymyn R. et al., Transformation of spin current by antiferromagnetic insulators, Phys. Rev. B 93, 224421 (2016).

This is especially important, since the idea of AFM birefringence and rotation of the spin wave polarization in the course of propagation in an AFM was explicitly introduced in [3].

In summary, this is an interesting and novel manuscript that can be published in the Nature Communication after a minor revision.

Responses to Referee Reports on manuscript NCOMMS-17-01280-T

Reply to remarks of the Referee #1

Magnonics, which aims to utilize SWs as information carriers, is currently one of extensively studied subjects in spintronics/magnetism communities as it offers low-power computing due to negligible Joule heating and nonlinear computing due to its wave characteristics. As described in the introduction, most magnonic devices up to now have focused on the control of SW amplitude or phase, but largely ignored the possibility to use the SW polarization. As proven in spintronics and photonics, the control of relevant polarization is essential for various applications. In this respect, the idea proposed in this work is attractive and very stimulating. The manuscript is well written so that even nonspecialists may get the point of how the proposed idea works. Also the proposed idea seems to work for synthetic antiferromagnets which are easy to fabricate in lab so that I expect this work will immediately motivate experimental groups.

On the overall, I find this work is very interesting and will attract considerable attention from magnonics/spintronics/magnetism communities. Thus I support the publication of this work in Nature Communications once the authors have a chance to consider the following minor comments.

Reply: We first thank the referee for the encouraging remarks regarding the quality and importance of the present work and also for his/her positive recommendation to *Nature Communications*.

At the end of the 1st paragraph in page 4, it is written that "... in-plane (out-of-plane) modes x-polarized (y-polarized) modes". It seems that x and y may need to be interchanged.

Reply: Thanks for the careful reading. We have corrected the text to "... in-plane (out-of-plane) modes y-polarized (x-polarized) modes ...", and also the text above to " \hat{e}_θ and \hat{e}_ϕ coincide with \hat{y} and \hat{x} ".

The authors mainly discuss the action of DW on SWs. It would be nice to discuss about the reaction: how the change in the SW polarization affects the DW? Is there any relevant magnonic torque enabling DW motion?

Reply: We thank the referee for raising this interesting question. The answer is YES, the inverse effect also exists, *i.e.* SW-driven domain wall motion depends on the polarization. Actually, we discovered this inverse effect first. The present manuscript only discusses the effect of using ‘rigid’ domain wall to manipulate the SW polarizations. We will have a separate manuscript soon discussing the inverse effect by allowing domain wall to move.

Reply to remarks of the Referee #2

Within the sine-Gordon model, which is usually used for FM and AF with uniaxial anisotropy, the domain wall does not scatter magnons (produces non-scattering potential for magnons). Any additional upgrading of the model (e.g., potential corresponding to tetragonal anisotropy) automatically gives rise to the scattering domain wall. If, in addition, this scattering is spin-dependent, then, the DW can have the properties of spin polarizer. Hence the idea is a clear one in principle, albeit not very innovative.

Reply: We first thank the referee for his/her critical comments, which help us a lot in improving our manuscript for better presentation.

It is general that any type of polarizer needs some sort of polarization (spin)-dependent scattering. This ‘principle’ is of course a clear one, nothing to innovate in this respect. However, how to realize such spin-dependent scattering is highly non-trivial, especially when it needs to be simple and efficient. In our manuscript, we proposed that a simple domain wall in AFM nanowires with DMI can naturally realize this goal. This idea of using magnetic texture itself, instead of any artificial physical structure, as magnonic ‘device’ is quite innovative in our opinion.

They start from LLG equations for each of magnetic sublattice and analyse excitations over the given equilibrium profile of the DW, analytically, by micromagnetic calc., and by what they claim to be a Green function technique.

Reply: We employ the Green function technique to calculate the transmission coefficients in our system because our magnonic transport system is a close analogy to the electronic transport system, where Green’s function is well developed and widely used. We have also checked that results from Green’s function method agree well with that from the mode-

matching method and the WKB method (not shown). We have modified our text about the Green function technique for better presentations in the revised manuscript. (Now extended and moved to the supplementary materials.)

There is however inconsistencies within their treatment. The authors consider the domain wall between two (equivalent? they do not explain) 180 domains. So, they implicitly assume that the ground state is homogeneous up/down long z axis. Moreover, as an equilibrium profile of the DW they use ansatz for AF without DMI.

The system with such a DMI is apt to formation of spiral and skyrmions. Homogeneous ground state (if any) can exist in a certain range of parameters which can be roughly estimated as follows. The system has two important lengthscales: the domain wall width $\sqrt{A/K}$ and a period of spiral A/D . Dimensionless parameter AK/D^2 separates two regimes. At $AK/D^2 \gg 1$ the ground state is almost homogeneous, and DMI modifies the DW and can (I am not sure) modify spectra. However, two homogeneous (up/down) states are equivalent in the limit of vanishingly small DMI.

At $AK/D^2 \ll 1$ the ground state is a spiral (or skyrmion). If there are two equivalent spiral states (I did not checked), then, the DW separates these states and its structure is different from the DW for 180 AF. In this case the author's approach is not valid at all.

The authors do not give any analysis of the parameters or validity of their model. However, at the end of the paper (Methods) give some numbers from which I deduce that they consider the most complicated case $AK/D^2=1$ in which the ground state is a spiral and spiral period is equal to the domain wall width. Analytical ansatz used for the DW profile is not valid.

Reply: We thank the referee to point out this extremely important point that we failed to mention in the previous manuscript.

In the simplest model, when the DMI is stronger than a critical value D_c , the ground state is indeed a spin spiral, which we have also confirmed, as shown in Fig. R1(a) below. In order to achieve relative strong spin-dependent scattering effects, we adopt a relative large value of D (i.e. $AK/D^2 \sim 1$), for which a simple domain wall would not be stable. In order to stabilize a domain wall at this large D , we used a free (or pinned) boundary condition (rather than the DM boundary condition) at the two ends of the AFM nanowire. As shown in Fig. R1(a), a domain wall structure is stabilized for such relatively large D in the micromagnetic simulations. Note that in all simulations, before injecting spin waves, we always relax the magnetic texture to its stable state first.

Fig. R1 **Micromagnetic simulations of static magnetic profiles with different boundary conditions.** The DMI strength is (a) $\tilde{D} = 3.0 \times 10^{-3} \text{ A} > \tilde{D}_c$ (the value used in the main text), domain wall configuration is stable for free and pinned boundary condition, and spin spiral is stable for DM boundary condition. (b) $\tilde{D} = 1.4 \times 10^{-3} \text{ A} < \tilde{D}_c$, domain wall configuration is always stable for all three types of boundary conditions. In both cases, the standard Walker profile is indicated by solid curves.

We would also like to emphasize that the presented polarizing and retarding effects in this manuscript exist for full range of DMI strength D . Smaller D values would result in similar polarizing and retarding effects, but less effective indeed. But this can be solved by chaining two or more domain walls together to enhance the effects, as shown in the supplementary materials.

As an example, we choose a smaller $\tilde{D} = D/\gamma = 1.4 \times 10^{-3} \text{ A} < \tilde{D}_c$, for which a domain wall is always stable regardless of the boundary condition as shown in Fig. R1(b). Fig. R2(a) shows the polarization-dependent spin wave scattering behaviors across the domain wall remain for this smaller D . And as confirmed by micromagnetic simulations in Fig. R2(b,c), a double-domain-wall structure can realize a very good spin wave polarizer at low frequency, and domain walls remain to function as spin wave retarders at high frequency.

We have now included separate supplementary materials discussing the effect of boundary condition on the stability of magnetic domain walls in AFM nanowire with DMI, and also the polarizing and retarding effects at smaller DMI strength D .

a. Polarization-dependent scattering

b. Polarizing effect with two domain walls

c. Retarding effect with four domain walls

Fig. R2 **Theoretical calculations and micromagnetic simulations of the polarizing and retarding effects at $\tilde{D} = 1.4 \times 10^{-3} \text{ A} < \tilde{D}_c$.** **a.** Theoretical calculations of the polarization-dependent scattering across antiferromagnetic domain walls at different DMI strengths D or across 2 consecutive domain walls. **b.** Micromagnetic simulation of the polarizing effect with 2 consecutive domain walls. **c.** Micromagnetic simulation of the retarding effect with 4 consecutive domain walls.

The authors also consider DMI as correction (which is not justified in the text) and assume that DMI "acts" only within the DW (because DMI is related with space derivative). In a homogenous phase DMI modifies magnons spectrum, which should include linear in k term. So, I cannot agree with the last statement in the paragraph below Eq. (3).

Reply: We do not consider DMI as correction. In fact, in our calculations DMI is present everywhere (except for boundary conditions). However, the effect of DMI only appears in the domain wall region where magnetization is varying in space.

As presented in Ref. [R1] and [R2], DMI would modify the magnon spectrum, but only when the spin wave wavevector has finite component in the direction of magnetization, *i.e.* when $\mathbf{k} \cdot \mathbf{m} \neq 0$, there would be linear \mathbf{k} term in the dispersion as mentioned by the referee. However, when the wavevector is perpendicular to the magnetization, the magnon

dispersion is not modified by DMI. This is the situation investigated in this manuscript, where \mathbf{m} is along $\hat{\mathbf{z}}$ and \mathbf{k} is along the nanowire axis $\hat{\mathbf{x}}$. Consequently, for the uniform domains in our structure, there is no linear \mathbf{k} term in dispersion as indicated by Eq. (3).

Moreover, in a homogenous state DMI modifies the profile of the DW, so, again, analytical ansatz used for the DW profile is approximate or not valid in the presence of DMI.

Reply: Whether the DMI would change the domain wall profile or not depends on direction of the easy-axis anisotropy, as shown in Fig. R3.

DMI would modify the domain wall profile when the easy-axis anisotropy (along \mathbf{x}) is along the wire direction as the one in Fig. R3(b), as also reported for an FM wire in Ref. [38]. But for the one under study in this manuscript, where the easy-axis (along \mathbf{z}) is perpendicular to the wire direction as in Fig. R3(a), the profile remains unchanged, as reported in Ref. [R3]. We confirmed this by micromagnetic simulations (see supplementary materials) that the domain wall profile is the standard Walker profile, even in the presence of DMI.

We have modified the manuscript to highlight the role of DMI in the domain wall profile in our model.

Fig. R3 **Schematics of domain wall profiles with DMI.** The easy-axis is (a) perpendicular to (b) along the wire. The magnetization profile is for \mathbf{m}_1 (red) and \mathbf{m}_2 (blue).

As a third comment, the authors consider DMI of the special type (inhomogeneous, Lifshits-like). The broken space-reversal symmetry allows not only inhomogeneous DMI, it can also allow homogeneous DMI. Homogeneous DMI means that the system is a weak AF, i.e., has

nonzero magnetization in the initial state. This also means that magnon modes with different spins are nondegenerate from the beginning, DW also has nonzero magnetization and this can influence substantially on the polarising properties of the DW. I understand, that homogenous DMI makes the model more complicated, but authors should explain explicitly why they disregard homogenous DMI.

Reply: Indeed, we only considered the inhomogeneous (Lifshits-type) DMI. This assumption is exact in artificial synthetic antiferromagnet. In real antiferromagnet, the homogeneous DMI may exist, and causes a misalignment of two opposite magnetizations in two magnetic sublattices, which induces weak ferromagnetism. However, the misalignment and ferromagnetism induced by the homogenous DMI is extremely small since it directly competes with the dominating antiferromagnetic coupling between two magnetic sublattices in AFM.

To estimate the misalignment, we introduce the effective magnetic field generated by homogenous DMI as $\mathbf{H}_{i,HD}^{eff} = D_H(-1)^i(\hat{\mathbf{x}} \times \mathbf{m}_i)$ with $\bar{1} = 2, \bar{2} = 1$, where D_H is the homogenous DMI strength, $\delta\theta$ is the tilt angle away from $\hat{\mathbf{z}}$, then we obtain $\delta\theta = \frac{1}{2} \text{atan} \frac{2D_H}{2J-K}$. Since $J \gg K, D_H$, $\delta\theta \ll 1$ thus the misalignment is extremely weak, and is expected to cause only negligible perturbations to the polarizing and retarding effects in our work.

Following the referee's suggestion, we have added a comment about why homogeneous DMI is omitted in our model in the revised manuscript.

Reply to remarks of the Referee #3

This is an interesting, novel and generally correct manuscript coming from one of the theoretical leaders in the field.

The authors propose to use polarization of spin waves in a dielectric antiferromagnetic (AFM) waveguide as a new way for information coding in beyond-CMOS AFM-based information processing systems, and demonstrate, that in the presence of DMI, an AFM domain wall can work as a spin wave polarizer or as a waveplate.

The main text of the manuscript is clearly written and will be accessible to a wide audience of the Nature Communications.

Reply: We thank the referee for the encouraging remarks regarding the quality and

importance of the present work and also for his/her positive recommendation to *Nature Communications*.

In the first sentence of the abstract the authors write : "...spin wave, or magnon when quantized, can propagate in both conducting and insulating materials without Joule heating." The text would become much clearer if the authors immediately mention that for spin wave Joule losses are simply replaced by losses associated with the Gilbert damping of propagating spin waves.

Reply: To avoid any confusion about dissipation, we have modified the text in the abstract, and also the description in the introduction correspondingly.

The authors, first state that they like AFM materials for polarization manipulation, because, in contrast with ferromagnets (FM), the time-reversal symmetry is restored in AFM. Then, they use AFM with DMI, where time-reversal symmetry is broken again by DMI. The general relation between the polarization manipulation for signal processing and the time-reversal symmetry should be explained in a correct and consistent way.

Reply: We thank the referee for pointing out the inconsistencies in the discussions based on the time-reversal symmetry.

We removed the time-reversal discussion in the revised manuscript. We now attribute the polarization freedom in AFM to the structure of two opposite magnetic sublattices. This is in contrast to FM, which only possesses one type of magnetic lattice, thus the polarization freedom is absent. Since DMI does not change the basic structure of two magnetic sublattices in AFM, the polarization freedom still holds.

The authors write in the introduction: "...polarization is an intrinsic property of wave-like (quasi-)particles such as light, acoustic wave, and spin wave." It seems to me, that "wave-like (quasi-)particles" are not "light, acoustic wave, and spin wave", but photons, phonons and magnons.

Reply: Thanks for careful reading. We have modified the statement to "...polarization is an intrinsic property of wave-like (quasi-)particles such as photons, phonons and magnons."

The referencing of the previous work in the field is inadequate. The authors should mention at least some of the work of their competitors in the same field:

1. Gomonay, H. V. & Loktev, V. M. *Spin transfer and current-induced switching in antiferromagnets. Physical Review B* 81, 144427 (2010).
 2. Gomonay, E. V. & Loktev, V. M. *Spintronics of antiferromagnetic systems (review article). Low Temp. Phys.* 40, 17–35 (2014).
 3. Khymyn R. et al., *Transformation of spin current by antiferromagnetic insulators, Phys. Rev. B* 93, 224421 (2016).
- This is especially important, since the idea of AFM birefringence and rotation of the spin wave polarization in the course of propagation an AFM was explicitly introduced in [3].*

Reply: We thank the referee for guiding us to these interesting and relevant papers. We have now included these relevant papers in the references [Refs. 26, 28 and 43] in the revised manuscript, and have specifically commented on the birefringence in the third paper by R. Khymyn.

References:

- [R1] Zhang, V. L. et al, In-plane angular dependence of the spin-wave nonreciprocity of an ultrathin film with Dzyaloshinskii-Moriya interaction. *Appl. Phys. Lett.* 107, 022402 (2015).
- [R2] Cortés-Ortuño D. and Landeros, P, Influence of the Dzyaloshinskii-Moriya interaction on the spin-wave spectra of thin films. *J. Phys.: Condens. Matter*, 25, 156001 (2013).
- [R3] Rohart, S. and Thiaville, A. Skyrmion confinement in ultrathin film nanostructures in the presence of Dzyaloshinskii-Moriya interaction. *Phys. Rev. B* 88, 184422 (2013)

Reviewers' Comments:

Reviewer #1 (Remarks to the Author):

I find the paper has been revised properly. I support the publication of this paper in Nature Communications.

Reviewer #2 (Remarks to the Author):

The authors have tried to address all the referee comments and have improved the paper significantly. However, the key weak point of the paper remains: the practical realisation of their idea. Though they claim that "All three boundary conditions can be constructed in an antiferromagnetic nanowire", they give no idea how to artificially create and keep 1800 domain walls in an antiferromagnet. In contrast to ferromagnets, 1800 (or translational) domains in the natural antiferromagnets are physically indistinguishable in the most of these materials. For this case, it is impossible to control the orientation of AF vector in a way necessary to successfully work as a polarizer.

On the other hand, in the synthetic antiferromagnets, the orientation of an AF vectors can be controlled.

I would suggest stressing that the presented idea can be realized ONLY in an synthetic AF or explain how to control the boundary conditions in the natural AFs.

I believe the paper can be accepted for publication with these changes.

Reviewer #3 (Remarks to the Author):

In the course of revision the authors took into account all the comments of the Referees. The revised manuscript can be published in the Nature Communications as is.

Responses to Referee Reports on manuscript NCOMMS-17-01280-A

Reply to remarks of the Referee #2

The authors have tried to address all the referee comments and have improved the paper significantly. However, the key weak point of the paper remains: the practical realisation of their idea. Though they claim that "All three boundary conditions can be constructed in an antiferromagnetic nanowire", they give no idea how to artificially create and keep 1800 domain walls in an antiferromagnet. In contrast to ferromagnets, 1800 (or translational) domains in the natural antiferromagnets are physically indistinguishable in the most of these materials. For this case, it is impossible to control the orientation of AF vector in a way necessary to successfully work as a polarizer. On the other hand, in the synthetics antiferromagnets, the orientation of an AF vectors can be controlled. I would suggest stressing that the presented idea can be realized ONLY in an synthetic AF or explain how to control the boundary conditions in the natural AFs. I believe the paper can be accepted for publication with these changes.

Reply: We first thank the referee's positive recommendation of our paper. We also would like to thank the referee's remarks on the previous version of our manuscript, which helped us greatly in improving our paper.

The antiferromagnetic domain wall has been observed by using the photoemission microscopy [R1] and spin-polarized scanning tunneling microscopy [R2] in natural antiferromagnetic materials. The controlling of AF vectors in natural antiferromagnets can be controlled by using exchange bias effect on the interface between FM and AFM materials [R3][R4][R5]. The AFM domain walls can be created, and also the boundary condition of the AFM wire can be engineered, based on such exchange bias effects [R6][R7]. Therefore, we would like to emphasize that the proposed polarizer and retarder in this manuscript work not only in synthetic antiferromagnets, but also in natural antiferromagnets.

We have now added some discussions correspondingly to our manuscript.

References:

- [R1] Weber, N. B., Magnetostrictive Domain Walls in Antiferromagnetic NiO. *Phys. Rev. Lett.* **91**, 237205 (2003).
- [R2] Bode, M., Atomic spin structure of antiferromagnetic domain walls. *Nat. Mat.* **5**, 447 (2006)
- [R3] Nolting, F. et al, Direct observation of the alignment of ferromagnetic spins by antiferromagnetic spins. *Nature* **405**, 767 (2000).
- [R4] Kim, J.-V. and Stamps, R. L., Hysteresis from antiferromagnet domain-wall processes in exchange-biased systems: Magnetic defects and thermal effects. *Phys. Rev. B* **71**, 094405 (2005).
- [R5] Mauri, D., Siegmann, H. C., Bagus, P. S. and Kay, E., Simple model for thin ferromagnetic films exchange coupled to an antiferromagnetic substrate. *J. Appl. Phys.* **62**, 3047 (1987).
- [R6] Cheng, R. and Niu, Q., Dynamics of antiferromagnets driven by spin current. *Physical Review B* **89**, 081105 (2014).
- [R7] Logan, J. M., et al, Antiferromagnetic domain wall engineering in chromium films. *Appl. Phys. Lett.* **100**, 192405 (2012).